# Consistency and Variation in Doublecortin and Ki67 Antigen Detection in the Brain Tissue of Different Mammals, including Humans

**DOI:** 10.3390/ijms24032514

**Published:** 2023-01-28

**Authors:** Marco Ghibaudi, Alessia Amenta, Miriam Agosti, Marco Riva, Jean-Marie Graïc, Francesco Bifari, Luca Bonfanti

**Affiliations:** 1Neuroscience Institute Cavalieri Ottolenghi (NICO), 10043 Orbassano, Italy; 2Department of Veterinary Sciences, University of Turin, 10095 Torino, Italy; 3Laboratory of Cell Metabolism and Regenerative Medicine, Department of Medical Biotechnology and Translational Medicine, University of Milan, 20133 Milan, Italy; 4Department of Biomedical Sciences, Humanitas University, 20090 Pieve Emanuele, Italy; 5IRCCS Humanitas Research Hospital, 20089 Rozzano, Italy; 6Department of Comparative Biomedicine and Food Science, University of Padova, 35020 Legnaro, Italy

**Keywords:** neurogenesis, immature neurons, comparative neuroplasticity, doublecortin, Ki67 antigen, mammalian brain

## Abstract

Recently, a population of “immature” neurons generated prenatally, retaining immaturity for long periods and finally integrating in adult circuits has been described in the cerebral cortex. Moreover, comparative studies revealed differences in occurrence/rate of different forms of neurogenic plasticity across mammals, the “immature” neurons prevailing in gyrencephalic species. To extend experimentation from laboratory mice to large-brained mammals, including humans, it is important to detect cell markers of neurogenic plasticity in brain tissues obtained from different procedures (e.g., post-mortem/intraoperative specimens vs. intracardiac perfusion). This variability overlaps with species-specific differences in antigen distribution or antibody species specificity, making it difficult for proper comparison. In this work, we detect the presence of doublecortin and Ki67 antigen, markers for neuronal immaturity and cell division, in six mammals characterized by widely different brain size. We tested seven commercial antibodies in four selected brain regions known to host immature neurons (paleocortex, neocortex) and newly born neurons (hippocampus, subventricular zone). In selected human brains, we confirmed the specificity of DCX antibody by performing co-staining with fluorescent probe for DCX mRNA. Our results indicate that, in spite of various types of fixations, most differences were due to the use of different antibodies and the existence of real interspecies variation.

## 1. Introduction

Developmental neurobiology is characterized by complex, highly dynamic processes that persist in the adult brain and allow extended assembly/modulation of the neural circuits (structural plasticity), including the addition of new neurons (adult neurogenesis). Recent progress in the field revealed increasingly complex landscapes involving different populations of “young”, undifferentiated neurons of different origin [1]. The new twists concern different populations of immature neuronal precursors belonging to canonical and non-canonical neurogenic processes, taking place both inside and outside of the neurogenic sites [2,3,4,5,6,7,8,9,10,11,12,13]. At least two populations of young, immature neurons coexist in the postnatal/adult brain: (i) newly born neurons generated from active division of adult neural stem cells, mainly hosted in the canonical neurogenic sites (the subventricular zone of the lateral ventricle and the dentate gyrus of the hippocampus) [3,5,6], and (ii) non-newly born “immature” or “dormant” neurons, which form during embryogenesis, then continuing to express markers of immaturity through adulthood [7,8,12,14,15,16,17]. The “dormant” neurons are located in brain regions not endowed with stem cell-driven neurogenesis, such as the cerebral cortex and amygdala [7,8,12,15,16,17]. These cell populations are not easy to distinguish since they share the same markers of immaturity during some phases of their life [10,11]. Such a distinction remained neglected for long time, leading to some misunderstandings in the interpretation of results and generating confusion in different types/sources of neurogenesis [1,11]. The origin of the young neurons can be revealed by the presence/absence of co-expression with markers of cell division, or pulse-chase experiments with 5-Bromo-2′-deoxyuridine (BrdU) and its analogues [18]. Yet, another important element of confusion consists of interspecies differences, with remarkable variation in the occurrence and distribution of the abovementioned types of plasticity and immature cell populations across mammals [7,19,20,21,22,23]. This fact makes it necessary to extend the level of investigation to large-brained, gyrencephalic species, or directly to humans [9,24,25,26]. When dealing with large-sized gyrencephalic brains some technical/practical difficulties arise, concerning fixation [27,28] and ethical issues [29,30] (Figure 1).

Consequently, most work must be performed on post-mortem brain tissues or intraoperative samples, thus putting limits to the experimental approach. For instance, the long-term tracing of the newly born elements (BrdU label-retaining cells) cannot be successfully used in humans or in mammalian species that are protected by international law (e.g., human primates and cetaceans). Alternatively, cell division can be detected by immunocytochemistry in post-mortem tissues, though limitedly to cells dividing at the exact time of animal death (e.g., Ki67 antigen [31]). Additionally, some discrepancies emerging after comparing the results with those obtained in laboratory rodents can be linked to real interspecific differences of the biological processes themselves [21,23,24,32,33,34,35] or to antibody specificity in the different mammals. Hence, the overall issue of analysing large, gyrencephalic brains is more complex than simply fixation, which overlap with natural interspecies variation and/or different specificity of antibodies across phylogeny, the latter two aspects being often neglected.

As shown in Figure 1, elements of complexity can be summarised as follows: (i) some markers of immaturity previously considered as specific for newly born neurons (e.g., doublecortin—DCX—and a polysialylated form of N-CAM—PSA-NCAM), are also expressed by wide populations of non-newly generated “immature” neurons [10,11]; (ii) both the rate of postnatal neurogenesis and the occurrence/distribution of immature neurons remarkably vary among mammals [20,21,23,36] (Figure 1B); (iii) when directly comparing brains widely different in size, belonging to animal species raising technical/ethical issues, some variables/approaches cannot be completely standardized (e.g., type and time of fixation, post-mortem interval; Figure 1C); (iv) the study of such variations require systematic, comparable approaches to actually match brain tissues belonging to widely different animal species, sizes and ages [23] (Figure 2).

Several antibodies from different manufacturers were employed over the years to detect the most popular antigens linked to structural plasticity/immaturity (e.g., DCX; Table 1) and cell division (e.g., Ki67 antigen; Table 2).

**Table 1 ijms-24-02514-t001:** Most common antibodies used to detect DCX in different animal species.

Host and Source	References	Animal Species (Common Name)
**Santa Cruz** ^1^ (goat)	Fasemore et al., 2018 [37]	Galago; Lemur; Potto
	Chawana et al., 2020 [38]	Egyptian fruit bat
	La Rosa et al., 2020 [23]	12 mammals (from mouse to chimpanzee)
	Kirby et al., 2012 [39]	Rat
	Fudge et al., 2012 [40]	Crab-eating macaque; southern pig-tailed macaque
	Zhang et al., 2009 [41]	Rhesus macaque
	Flor-Garcìa et al., 2020 [42]	Human
	Sorrells et al., 2018 [24]	Human; rhesus macaque
	Tobin et al., 2019 [43]	Human
	Boekhoorn et al., 2006 [44]	Human
	Liu et al., 2020 [45]	Rhesus macaque
	Marlatt et al., 2011 [46]	Common marmoset
	Parolisi et al., 2017 [32]	Dolphin
	La Rosa et al., 2018 [47]	Dolphin; sheep
	Moreno-Jiménez et al., 2019 [25]	Human
	Jin et al., 2006 [48]	Human
	Crews et al., 2010 [49]	Human; mouse
	Knoth et al., 2010 [50]	Human
	Wang et al., 2011 [51]	Human; rhesus macaque
	Gomez-Nicola et al., 2014 [52]	Human; mice
	Ekonomou et al., 2015 [53]	Human
	Dennis et al., 2016 [54]	Human
	Galàn et al., 2017 [55]	Human
	Liu et al., 2008 [56]	Human
	Ponti et al., 2006 [57]	Rabbit
	Kunze et al., 2015 [58]	Mouse
	Cai et al., 2009 [59]	Human; rhesus macaque; cat
	Verwer et al., 2007 [60]	Human
	Bloch et al., 2011 [61]	Human; cynomolgus monkey; African green monkey
	Li et al., 2022 [62]	Human
**Abcam** (rabbit)	Piumatti et al., 2018 [7]	Sheep
	Jhaveri et al., 2018 [63]	Mouse
	Flor-Garcìa et al., 2020 [42]	Human
	Sorrells et al., 2018 [24]	Human; rhesus macaque
	Cipriani et al., 2018 [64]	Human
	Tobin et al., 2019 [43]	Human
	Liu et al., 2020 [45]	Rhesus macaque
	Parolisi et al., 2017 [32]	Dolphin
	La Rosa et al., 2018 [47]	Dolphin; sheep
	Wang et al., 2011 [51]	Human; rhesus macaque
	Nogueira et al., 2014 [65]	Human
	Perry et al., 2012 [66]	Human
	Bloch et al., 2011 [61]	Human; cynomolgus monkey; African green monkey
	Cai et al., 2009 [59]	Human; rhesus macaque; cat
**Millipore** (guinea pig)	Akter et al., 2020 [67]	Common marmoset
	Benedetti et al., 2019 [68]	Mouse
	Jhaveri et al., 2018 [63]	Mouse
	Flor-Garcìa et al., 2020 [42]	Human
	Sorrells et al., 2018 [24]	Human; rhesus macaque
	Sorrells et al., 2019 [9]	Human
	Cipriani et al., 2018 [64]	Human
	Gomez-Nicola et al., 2014 [52]	Human; mice
	Paredes et al., 2016 [69]	Human
	Kunze et al., 2015 [58]	Mouse
	Bloch et al., 2011 [61]	Human; cynomolgus monkey; African green monkey
	Alderman et al., 2022 [17]	Human; mouse
**Cell Signalling** **Technology** (rabbit)	Sorrells et al., 2018 [24]	Human; rhesus macaque
	Liu et al., 2008 [56]	Human
	Maheu et al., 2015 [70]	Human
	Paredes et al., 2016 [69]	Human
	Sorrells et al., 2019 [9]	Human
	Martì-Mengual et al., 2013 [71]	Human; squirrel monkey; cat
	Alderman et al., 2022 [17]	Human; mouse
	Coviello et al., 2022 [72]	Human

^1^ out of production (stock in the Turin lab).

**Table 2 ijms-24-02514-t002:** Most common antibodies used to detect Ki67 antigen in different species.

Host and Source	References	Animal Species
**Leica-Novocastra** ^1^ (rabbit)	Fasemore et al., 2018 [37]	Galago; lemur; potto
	Akter et al., 2020 [67]	Common marmoset
	Chawana et al., 2020 [38]	Egyptian fruit bat
	La Rosa et al., 2020 [23]	12 mammals (from mouse to chimpanzee)
	Jhaveri et al., 2018 [63]	Mouse
	Sorrells et al., 2018 [24]	Human; rhesus macaque
	Sorrells et al., 2019 [9]	Human
	Tobin et al., 2019 [43]	Human
	Boekhoorn et al., 2006 [44]	Human
	Quiñones-Hinojosa et al., 2006 [73]	Human
	Fahrner et al., 2007 [74]	Human
	Parolisi et al., 2017 [32]	Dolphin
	La Rosa et al., 2018 [47]	Dolphin; sheep
	Martì-Mengual et al., 2013 [71]	Human; squirrel monkey; cat
**BD Pharmingen** (mouse)	La Rosa et al., 2020 [23]	12 mammals (from mouse to chimpanzee)
	Sorrells et al., 2018 [24]	Human; rhesus macaque
	Sorrells et al., 2019 [9]	Human
	La Rosa et al., 2018 [47]	Dolphin; sheep
**Abcam** (rabbit)	Gomez-Nicola et al., 2014 [52]	Human; mice
	Allen et al., 2016 [75]	Human
	Cipriani et al., 2018 [64]	Human

^1^ out of production.

Therefore, heterogeneous results were reported by different authors by using various methods and antibodies in each species, in a manner that makes it difficult to really compare data [23,76]. Although focus has been put on quantification methods [76], real interspecies differences in the occurrence/distribution of antigens as well as antibody specificity can play a role. Here, we tried to combine some of the abovementioned variables in six mammals, including humans (Figure 2), with a twofold aim: (i) to map the best performance of antibodies raised against the most used markers for neuronal immaturity (DCX) and cell division (Ki67 antigen) in each species, and (ii) to reach a comparable landscape for these markers across species. The screening was performed on four selected brain regions in which the occurrence of DCX and Ki67 staining is well known, two of them hosting non-newly generated “immature” neurons (paleocortex and neocortex) and the other two hosting newly born neurons (hippocampus and subventricular zone) (Figure 2 and Figure 3). The study was conceived on two levels: (a) detection of DCX and Ki67 antigen immunoreactivity by testing seven commercial antibodies in five mammals (including lissencephalic and gyrencephalic species), by considering the abovementioned brain regions (Figure 3); (b) detection of DCX in the human brain cerebral cortex in combination with its mRNA by using the RNAscope technique (Figure 3).

In spite of an obvious interest in visualising DCX in the human brain, the detection of this antigen in the cerebral cortex of primates has been controversial, spanning from claiming its occurrence in most cortical layers [61] to its very low level due to non-specific staining [45]. Due to the complexity of the approach (seven antibodies tested in four neuroanatomical regions obtained from 24 brains of six widely different species) only qualitative aspects were considered, including cases of occurrence/absence of staining, background, or obvious non-specific staining (Figure 4).

## 2. Results

As shown in Figure 4, we focused on the quality of staining defined by four parameters, spanning from a clear and clean staining of the specific cell populations to the absence of immunocytochemical signal. Results obtained from the comparative analysis in mouse, marmoset, rabbit, cat and sheep are reported in Figure 4, Figure 5, Figure 6 and Figure 7 and Table 3 and Table 4. Full results obtained with and without antigen retrieval are reported in Table 3 and Table 4, whereas confocal images are provided for citrate treatment only (Figure 5a,b and Figure 6). The set of results concerning the detection of DCX in the human cerebral cortex is reported in Table 5 and Figure 8.

### 2.1. Comparative Immunostaining in the Brain of Five Mammals

Tissues analysed in this study belong to widely different mammals endowed with different brain size and gyrencephaly (Figure 2 and Figure 3); for this reason, they did not undergo the same type of fixation (Table 6). We tried to obtain maximal homogeneity for heterogeneous tissues, in terms of fixation and post-mortem interval, as previously reported in a study involving 12 mammalian species [23]. Nevertheless (as described in the Introduction, and also in view of the specific aims of the present study), we included specimens fixed by perfusion (mouse, rabbit, sheep), immersion (cat, marmoset, sheep, human), post-mortem tissues and intraoperative samples (human). Considering sheep, both perfused and immersed brains were collected, in order to extend our comparison. As a result, a substantial ability to detect the two antigens in all animal species was observed (out of 115 immunocytochemical staining samples performed for DCX with 5 min (5′) citrate treatment, 89 showed a positive signal while only 26 showed no signal or non-specific staining; see Table 3a), with no difference among the three specimens analysed in each species. Nevertheless, substantial differences in the occurrence/type of staining were also observed (purple areas in Table 3b and Table 4b). By observing the distribution of the staining rates (success and failure, the latter including the absence of signal and a non-specific staining), they are not strictly linked to the animal species or the type of fixation, but rather to the use of different antibodies (in some cases, also to real interspecies differences; Figure 7A).

**Table 3 ijms-24-02514-t003:** (**a**) Occurrence and quality of staining for DCX. (**b**) Merge (with and without citrate treatment).

(**a**)							
**Citrate Treatment (5 min)**							
**Antibodies**	**Regions**	**Mouse**	**Marmoset**	**Rabbit**	**Cat**	**Sheep** (perfusion)	**Sheep** (immersion)
**Santa Cruz** (goat)	**pc**						** ● **
	**nc**						** ● **
	**svz**						●
	**sgz**					*****	** ● **
**Cell Signalling** (rabbit)	**pc**			*****	*****	*****	*****
	**nc**		*****	*****	*****	*****	
	**svz**		*****				
	**sgz**		*****	*****	*****	*****	*****
**Abcam** (rabbit)	**pc**	*****	** ● **	*****	*****	*****	*****
	**nc**		** ● **	*****	*****	*****	
	**svz**		●				
	**sgz**	*****	●		*****	*****	*****
**Santa Cruz** (mouse)	**pc**	*****		*****		*****	** ● **
	**nc**			*****	*****	*****	** ● **
	**svz**		*****				** ● **
	**sgz**		*****	*****			** ● **
**Millipore** (guinea pig)	**pc**	*****		*****	*****	*****	
	**nc**			*****	*****	*****	
	**svz**				*****	*****	
	**sgz**	*****		*****	*****	*****	*****
**No Citrate**							
**Antibodies**	**Regions**	**Mouse**	**Marmoset**	**Rabbit**	**Cat**	**Sheep** (perfusion)	**Sheep** (immersion)
**Santa Cruz** (goat)	**pc**				*****	*****	** ● **
	**nc**		*****		*****	*****	** ● **
	**svz**					*****	●
	**sgz**		*****		*****	*****	** ● **
**Cell Signalling** (rabbit)	**pc**			*****	*****	*****	** ● **
	**nc**		*****	*****	*****	*****	** ● **
	**svz**		*****	*****			** ● **
	**sgz**		*****	*****	*****	*****	** ● **
**Abcam** (rabbit)	**pc**	*****	** ● **	*****	*****	*****	
	**nc**		●	*****	*****	*****	
	**svz**		●				
	**sgz**		●	*****	*****	*****	*****
**Santa Cruz** (mouse)	**pc**	*****	*****	*****	*****		
	**nc**		*****	*****	*****		** ● **
	**svz**	*****	*****	*****	*****		
	**sgz**	*****	*****	*****	*****		
**Millipore** (guinea pig)	**pc**	*****	*****	*****	*****	*****	
	**nc**		*****	*****	*****	*****	
	**svz**	*****	*****	*****	*****		
	**sgz**	*****	** ● **	** ● **	*****	*****	
(**b**)							
**Antibodies**	**Regions**	**Mouse**	**Marmoset**	**Rabbit**	**Cat**	**Sheep** (perfusion)	**Sheep** (immersion)
**Santa Cruz ** (goat)	**pc**				**(5′)**	**(5′)**	
	**nc**		**(5′)**		**(5′)**	**(5′)**	
	**svz**					**(5′)**	
	**sgz**		**(5′)**		**(5′)**		
**Cell Signalling** (rabbit)	**pc**						**only 5′ C**
	**nc**						**only 5′ C**
	**svz**		**(5′)**	**(5′)**			**only 5′ C**
	**sgz**		**(5′)**				**only 5′ C**
**Abcam** (rabbit)	**pc**			**without C**			**without C**
	**nc**		**without C**				
	**svz**						
	**sgz**	**without C**		**(5′)**			
**Santa Cruz ** (mouse)	**pc**	**only 5′ C**		**only 5′ C**	**only 5′ C**	**only 5′ C**	
	**nc**			**only 5′ C**		**only 5′ C**	
	**svz**		**only 5′ C**	**(5′)**	**(5′)**	**only 5′ C**	
	**sgz**	**(5′)**	**only 5′ C**		**only 5′ C**		
**Millipore** (guinea pig)	**pc**	**(5′)**					
	**nc**						
	**svz**	**(5′)**		**(5′)**		**without C**	
	**sgz**					**without C**	

Subtable (**a**): 
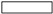
 Absence of staining due to real interspecies difference (e.g., no cell population containg DCX in the mouse neocortex); 
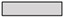
 Clear and clean immunoreaction; 
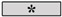
 Immunoreaction with background; 
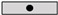
 Immunoreaction with non-specific staining; 
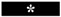
 No signal with background; 
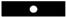
 No signal with non-specific staining; 
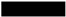
 No signal; Subtable (**b**): MERGE of results obtained with and without 5′ Citrate: 
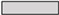
 Same results; 
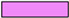
 Different results; **(5′)** better with Citrate; **only (5′)** C signal only with Citrate; **without C** signal only without Citrate.

**Figure 5 ijms-24-02514-f005:**
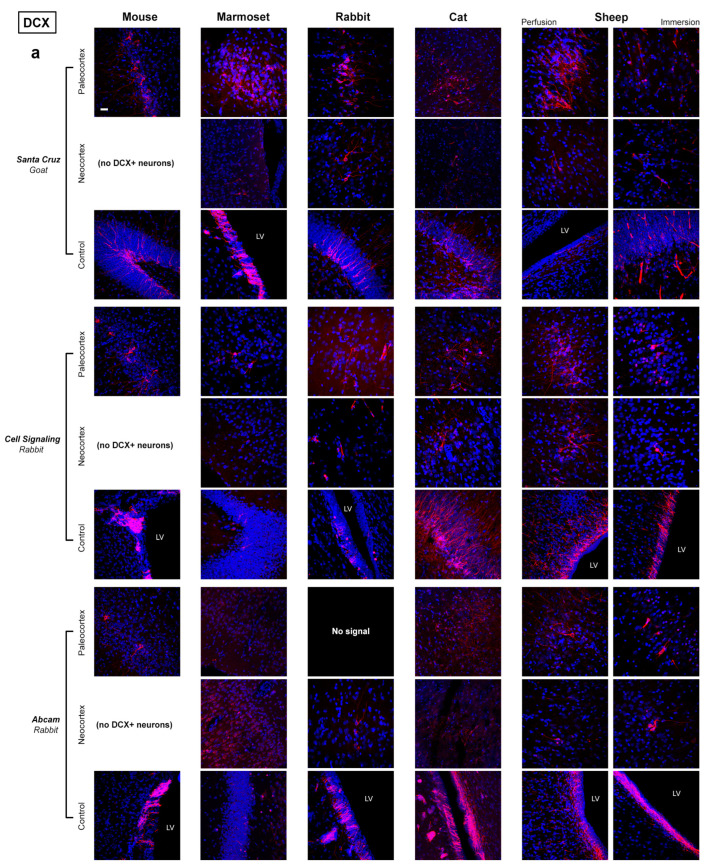
(**a**,**b**). Representative confocal images of DCX detection (in red) in different mammals for five different antibodies. All specimens are counterstained with DAPI. All photographs were taken at the same magnification (scale bar: 30 µm). Control is represented by one of the two neurogenic sites, either SVZ or hippocampus. LV: lateral ventricle.

### 2.2. Regional Differences

Some recurrent regional differences depending on the neuroanatomical area investigated were observed for DCX. A clear and clean staining was generally detectable in the SVZ (21 positivities out of 30, with 8 cases of background or non-specificity), whereas the occurrence of background or non-specific staining was more frequent in the SGZ (15) and cerebral cortex layer II (23). These differences, observed in different brain regions of the same animal species, do not seem linked to antibody specificity, rather to neuroanatomical features and to the cell populations involved (see also Section 3). Moreover, the impact of the total staining in the microscope field can be different depending on the structures detected: the SVZ hosts masses of neuroblast-forming chains enriched in DCX, whereas isolated neurons at different maturational stages are detectable in the dentate gyrus, even more diluted in space and less immature in the neocortex.

The detection of Ki67 antigen was more homogeneous with respect to the region investigated, mainly because the analysis was restricted to neurogenic sites (Figure 6 and Table 4). The rate of failure and the occurrence of background noise were slightly constant in both regions. Being Ki67 a nuclear antigen, can be hardly reached by antibodies in tissue fixed by immersion; for this reason, it requires longer citrate treatment to unmask antigenicity (Figure 6).

**Figure 6 ijms-24-02514-f006:**
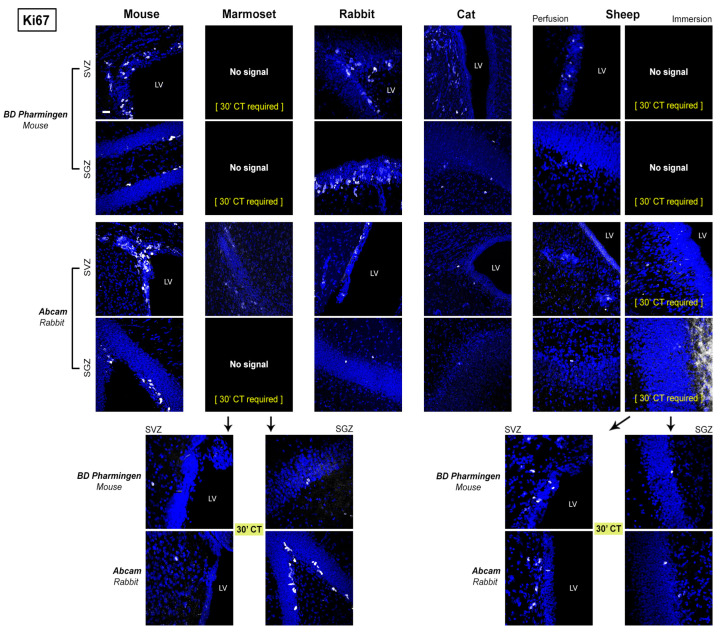
Representative confocal images of Ki67 antigen detection (in white) in different mammals for two different antibodies in the two neurogenic sites (forebrain SVZ and hippocampal SGZ). All specimens are counterstained with DAPI. All photographs have been performed at the same magnification (scale bar: 30 µm). In marmoset, (30′ CT) indicates that the antigen can be detected only after 30 min of citrate buffer treatment. LV: lateral ventricle.

**Table 4 ijms-24-02514-t004:** (**a**) Occurrence and quality of staining for Ki67 antigen. (**b**) Merge (with and without citrate treatment).

(**a**)							
**Citrate Treatment (5 min)**							
**Antibodies**	**Regions**	**Mouse**	**Marmoset**	**Rabbit**	**Cat**	**Sheep** (perfusion)	**Sheep** (immersion)
**BD Pharmingen** (mouse)	**svz**		*****				*****
	**sgz**						*****
**Abcam** (rabbit)	**svz**		*****		*****	*****	*****
	**sgz**		*****		*****	*****	●
**No Citrate**							
**Antibodies**	**Regions**	**Mouse**	**Marmoset**	**Rabbit**	**Cat**	(perfusion)	(immersion)
**BD Pharmingen** (mouse)	**svz**	*****	*****		*****		*****
	**sgz**	** ● **	*****		*****		** ● **
**Abcam** (rabbit)	**svz**		*****	●	*****	*****	*****
	**sgz**		*****			*****	*****
(**b**)							
**Antibodies**	**Regions**	**Mouse**	**Marmoset**	**Rabbit**	**Cat**	**Sheep** (perfusion)	**Sheep** (immersion)
**BD Pharmingen** (mouse)	**svz**	**(5′)**			**(5′)**	**only 5′ C**	
	**sgz**	**only 5′ C**			**only 5′ C**	**only 5′ C**	
**Abcam** (rabbit)	**svz**		**only 5′ C**	**(5′)**			**only 5′ C**
	**sgz**			**only 5′ C**	**only 5′ C**		**only 5′ C**

Subtable (**a**): 
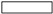
 Absence of staining due to real interspecies difference (e.g., no cell population containg DCX in the mouse neocortex); 
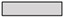
 Clear and clean immunoreaction; 
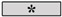
 Immunoreaction with background; 
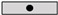
 Immunoreaction with non-specific staining; 
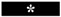
 No signal with background; 
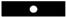
 No signal with non-specific staining; 
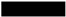
 No signal; Subtable (**b**): MERGE of results obtained with and without 5′ Citrate: 
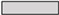
 Same results; 
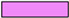
 Different results; **(5′)** better with Citrate; **only (5′)** C signal only with Citrate; **without C** signal only without Citrate.

**Figure 7 ijms-24-02514-f007:**
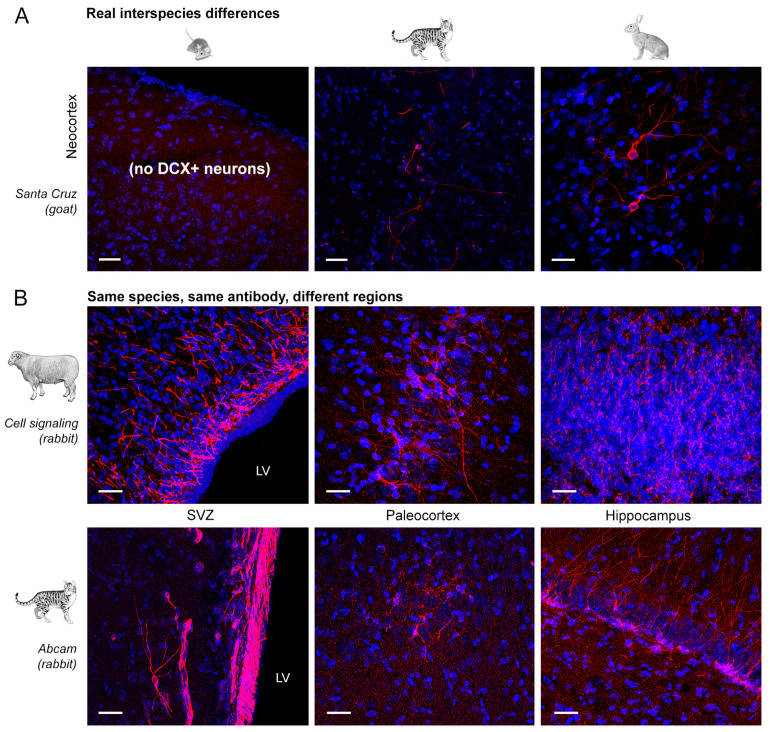
Variation in the type and quality of immunostaining obtained in different animal species (**A**) and in different brain regions (**B**). (**A**) In comparative studies the detection DCX+ cell populations can yield different results depending on real interspecies differences, e.g., the absence of DCX+, immature neurons in the mouse neocortex [23]. (**B**) Even in the same species, using the same antibody, different results can be found depending on the brain region. In most cases, the SVZ bordering the lateral ventricle (LV) stains far more clean and clear with respect to parenchymal regions such as the cortex and the hippocampus (sheep and cat). Note that in sheep, SVZ and cortex stain for the expected neuronal populations and non-specific staining is detectable in the hippocampus. Scale bar: 30 µm.

### 2.3. DCX Detection in the Human Cerebral Cortex

In the human cerebral cortex, the picture appeared different from that observed in the other mammals, since only one antibody was properly working. Results obtained in humans are reported in Table 5, Figure 8 and Figure 9. Analyses were restricted to the neocortex, both from post-mortem tissues and intraoperative samples, in the latter integrated with RNAscope analysis in order to assess the spatial expression of RNA molecules with cellular specificity for DCX (Figure 9).

**Table 5 ijms-24-02514-t005:** Occurrence and quality of staining for DCX in human brain tissue.

Antibody	Region	Immuno	RNAscope
**Santa Cruz** (goat)	Cerebral cortex(temporal)	●	-
**Cell Signalling** (rabbit)	●	-
**Abcam** (rabbit)	*	Co-expression in some DCX+ neurons in layer II-III
**Santa Cruz** (mouse)	●	No co-expression in DCX+ neurons (non-specific staining)
**Millipore** (guinea pig)	●	No co-expression in DCX+ neurons (non-specific staining)

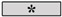
 Immunoreaction with background; 
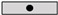
 Immunoreaction with non-specific staining; 
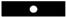
 No signal with non-specific staining.

All antibodies raised against DCX and used on the other mammals were tested immunocytochemically on coronal cryostat sections of human temporal cortex (Figure 8A). Among the five antibodies employed, only the Abcam rabbit and the Millipore guinea pig revealed neuronal-like cells (mostly localised in the superficial cortical layers II and III), the other antibodies giving no signal and/or a non-specific punctate reaction (Table 5 and Figure 8A). The detection of DCX+ neuronal-like cells was rare and revealed elongated cell bodies whose staining extended only in a short part of a process, as expected in human brains of old individuals (ranging from 67 to 81 years in our study; Table 7 [59,62,72,77]). Other cell bodies located in different layers and showing a faint staining resembling autofluorescence were also observed. To check whether the immunostaining was specifically associated with DCX-expressing cells, an in situ hybridization with RNA probe (RNAscope) was performed in double staining with Abcam, Millipore and Santa Cruz anti-DCX antibodies (Figure 9). Only in the case of Abcam antibody a co-expression was detected, revealing a subpopulation of DCX-expressing neurons in layers II and III (Figure 9 and Table 5).

Post-mortem, heavily formalin-fixed human tissues were also used, to make a comparison. In these tissues, by using the Abcam antibody some unipolar/bipolar neurons were detectable in the cortical layer II (Figure 8B, top); yet, interlaminar astrocytes of the layer I [78] were also heavily stained (Figure 8B, bottom). To further validate DCX expression in these cells, we combined RNAscope analysis for DCX gene expression with immunocytochemistry for the astrocytic marker GFAP and found no GFAP+ astrocytes co-expressing DCX mRNA (Figure 9). This result, together with the absence GFAP+/DCX+ detection in intraoperative samples, indicate that GFAP+/DCX+ double positive cells detected in the heavily formalin-fixed human tissues are likely due to non-specific staining.

**Figure 8 ijms-24-02514-f008:**
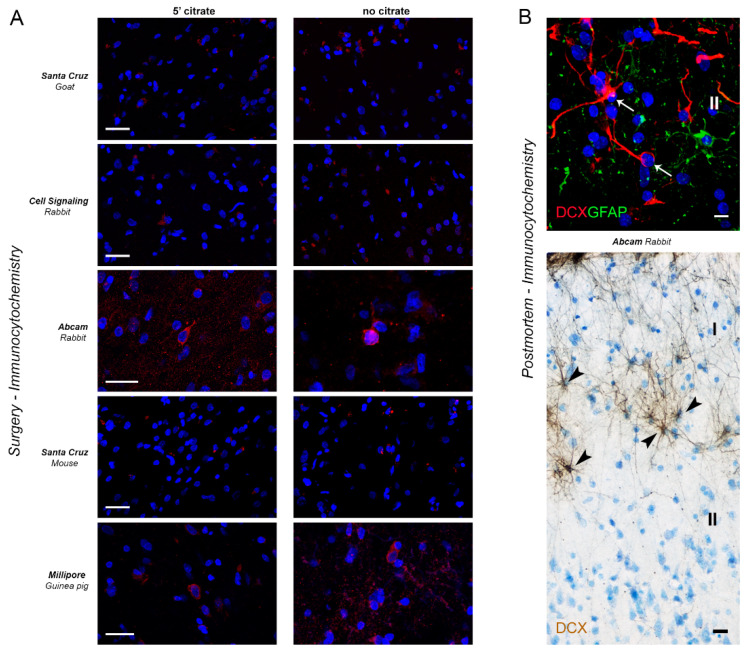
Detection of DCX in the human neocortex by immunocytochemistry. (**A**) Immunocytochemical staining for DCX (red) by using different antibodies, both with and without citrate buffer treatment. Confocal images taken in cortical layer II-III (I, II, cortical layers). (**B**) Immunocytchemistry on post-mortem brain tissue with Abcam primary antibody reveals some unipolar/bipolar neurons (arrows) in layer II, but also a non-specific staining on interlaminar astrocytes in layer I (arrowheads; see Figure 9). Scale bars: 20 µm.

**Figure 9 ijms-24-02514-f009:**
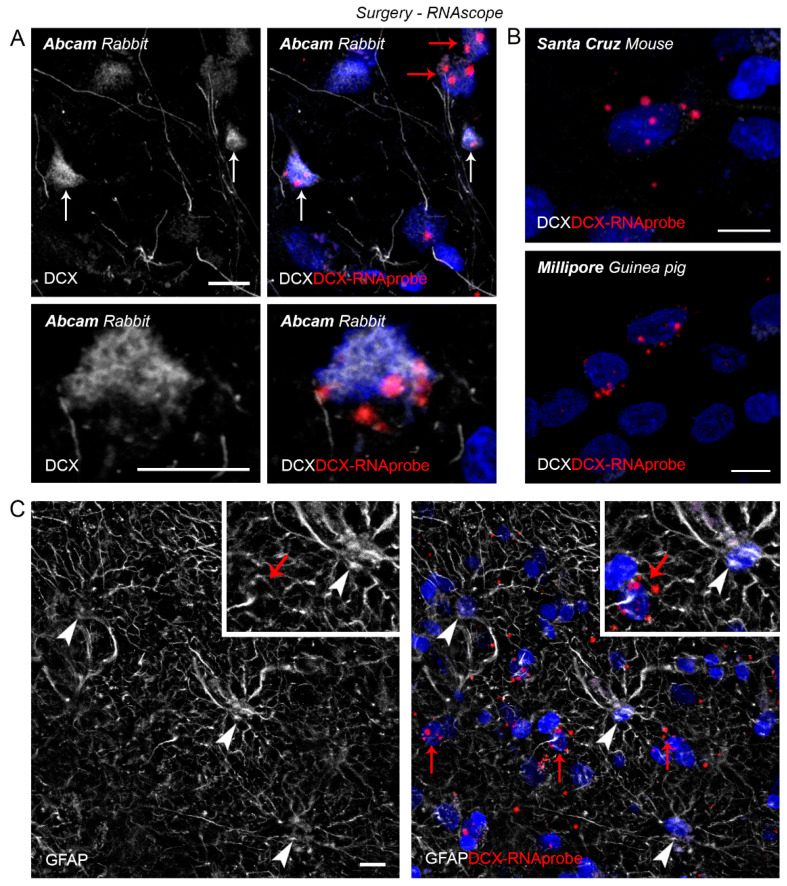
Detection of DCX in the human neocortex by RNAscope. (**A**) RNAscope technique in association with anti-DCX Abcam primary antibody confirms that some labelled cells in layer II-III actually are DCX-expressing cells (white arrows); some cells positive only for the RNA-probe do not express the protein (red arrows); bottom, higher magnification of a doule-labelled cell. (**B**) No co-expression of DCX-RNAprobe (red arrows) with anti-DCX Santa Cruz and Millipore. (**C**), No co-expression of DCX-RNAprobe (red arrows) with GFAP+ astrocytes (white arrowheads) was detected. Scale bars: 10 µm.

## 3. Discussion

Cell marker immunocytochemical detection is an important tool in developmental neurobiology, helping to define different cell populations on the basis of their phenotype and/or maturational stages. The reliability of antibodies used to detect such markers is obviously a prominent aspect and acquires special complexity in comparative studies involving widely different mammalian species characterized by different brain size and inherent difficulties in obtaining fresh, well-fixed material (Figure 1). When protected animal species or humans are involved, the post-mortem interval, as well as the type of fixation, cannot be the same as in laboratory rodents. Consequently, even when several animal species are considered in the same study and thus processed by using the same methods, the original conditions of the tissues cannot be exactly the same [23,79,80,81]. Conversely, when a single animal species is investigated to find the best conditions to detect antigens in that brain tissue (a most frequent event in the literature), comparison with other species can be tricky.

On these bases, we performed here a systematic testing of different commercial antibodies raised against two widely used markers of structural plasticity on different brain regions (neurogenic and non-neurogenic) of five mammalian species and on the cerebral cortex of humans, in search of similarities and possible substantial, qualitative differences. The first aim was to establish a screening on the most commonly used antibodies to check whether they can be considered specific for all species, or otherwise, to map which antibodies do not work in some of them, in order to obtain a panel to be used in future comparative studies, with the aim of reaching a “comparable” picture.

### 3.1. Variables Affecting the Occurrence/Quality of Staining

The present study involved seven antibodies raised against two antigens, detected in four brain regions of five mammalian species, as well as in the cerebral cortex of humans (Figure 2). Our results revealed that, in addition to a substantial prevalence of positive immunostainings, remarkable differences can exist when performing comparative analyses in mammals endowed with widely different brains (Figure 4, Figure 5, Figure 6, Figure 7, Figure 8 and Figure 9). In general, by considering the data obtained here and those collected from the current literature (Table 1 and Table 2), the cytoskeletal protein DCX and Ki67 antigen appear to be well conserved through phylogeny and particularly resistant to fixation. As an extreme example, both antigens were detectable in internal positive controls of dolphin cerebella (external germinal layer at early postnatal stages) that were collected with a relatively long post-mortem delay (varying between 18 and 40 h) and kept in fixative (4% buffered formalin) for months, even years [32]. Nevertheless, variation in the occurrence and/or quality of staining is frequently reported in comparative studies (references in Table 1 and Table 2). Accordingly, while the detection of the two antigens was mostly successful in the present study, we also showed some remarkably different results depending on the animal species investigated and the antibody employed. The most significant evidence emerging from our comparative analysis is the finding that DCX and Ki67 antigen can be successfully detected by using different antibodies in different animal species, in spite of various types/degrees of fixation of the different brains studied (Figure 10). Hence, taking for granted that an appropriate tissue fixation is always required, the next step in comparative studies should be to find the right antibody(ies) tailored for the species under investigation. Our results indicate that the right antibodies can be successful in a relatively wide range of different types of fixation (Figure 10).

To a lesser extent, differences were observed depending on the brain region investigated. Regional neuroanatomy can affect the quality of fixation and staining. The SVZ, lining the ventricular cavity, can be easily reached by the fixative, especially when using a tissue immersion procedure; then, SVZ and SGZ neurogenic sites are enriched in neuroblasts, which are filled in with DCX [82], whereas some of the “immature” neurons of the cortical layer II can show a lower content of DCX protein, due to their state of immature cells (thus, in a more advanced maturational stage than neuroblasts; see for example [83]). This seems to be the case of cortical immature neurons in the brain of old animals [59,84] and adult/old human individuals [62,77]. It is worth noting that in some cases, a different staining (e.g., with or without background) can be obtained in different brain regions in spite the same antibody dilution is used (see Section 2).

In some cases, when dealing with comparison between species, the absence of staining can be simply due to interspecies differences. We now know that some DCX+ cell populations can be present or not in different species depending on the brain region or animal age. Here we show the example of the layer II cortical “immature” neurons, which are present in the neocortex of non-rodent mammals, being absent in mice (Figure 5 and Figure 7). This can appear trivial, yet the wide use of laboratory rodents as an almost exclusive animal model for biomedical research has induced many scientists to expect that results coming from other animal species should replicate what previously shown in laboratory mice [85]. Comparative studies are revealing that remarkable differences can exist concerning the occurrence/rate/location of various forms of brain structural plasticity, especially regarding the different origin of the DCX+ “young” neurons [1,10]. While differences/similarities concerning stem cell-driven adult neurogenesis in rodents and humans is currently controversial [27,28], substantial interspecies differences are known to exist for the non-newly born, immature neurons of the cerebral cortex [1,7,23]. For these reasons, a more complete mapping of structural plasticity processes across mammals, brain regions and ages is needed. This can be accomplished by addressing the entire issue of interspecies differences, tissue collection/fixation, and antibody specificity considering that in some cases results can be different from what has been established in rodents.

### 3.2. Detecting DCX in the Human Cerebral Cortex

In humans, this study was restricted to the neocortex, a region of uttermost importance in cognition [86], as well as in pathology [87], which has been recently shown to host a population of “immature” DCX+ neurons [62,72], substantially absent in mice [23]. DCX detection in the human cerebral cortex, and more in general in the cortex of primates, has been controversial for a long time. For instance, some authors reported a widespread occurrence of this cytoskeletal protein in most cortical layers of the macaque cerebral cortex [61], while in a recent report, carried out on 9–10-year-old macaques with Western blot analysis, immunocytochemistry and antigen adsorption controls [45], it was concluded that some antibodies might show non-specific staining, its presence resulting in far more spatial restriction. For this reason, our analysis in the human cortex was extended to an RNAscope assay to check the visualization of RNA molecules in individual cells, thus revealing gene expression beyond the simple protein transcription. Our results substantially confirm the Liu et al. findings, since only one of the five antibodies tested by immunocytochemistry (the Abcam antibody) did produce a staining co-expressing with the RNAprobe (Figure 8 and Figure 9) and was restricted to a small population of cortical layer II-III neurons.

### 3.3. Comparative Antibody Performance

One of the main problems in comparative research is the absence of systematic, comparable analyses among widely different mammals. Most studies on brain plasticity and neurogenesis were carried out on very standardized animal models, such as the laboratory rodents [85]. As a result, the gap between our knowledge in rodents and humans remains widely unexplored, hampering the identification of real phylogenetic variations that might be useful for the correct interpretation of data obtained in mice and for a proper translation [85,88]. Nonetheless, most comparative studies consider either a single animal species or a small group of them, while analysing different species with the same method/approach in a study is rare (e.g., [23,79,80]). Finally, in studies using immunocytochemical markers on post-mortem tissues, several antibodies raised against the same antigen are often reported in the Section 4 without stating which of them achieved successful staining or in which condition/treatment.

A clear result emerging from the present comparative study is the lack of an “optimal” antibody for all the animal species considered. Overall, there is not an animal species among those considered (apart from mice) in which all antibodies are working (Table 3, Table 4, Table 5 and Table 6). On the other hand, the antigens were always detectable in each species with one or more antibodies (what can be considered as further indication of specific staining). Those providing the best results in each species are reported in Figure 10 below tables for DCX and Ki67. Especially for DCX, the association between animal species and successful staining appears to shift from mouse to humans, indicating that substantial differences must be considered along phylogeny. Of importance, most of the tested antibodies worked well in small-brained mammals whereas the possibility for successful, specific staining progressively decreased towards humans. As reported above, the type of fixation did not impact a lot in the whole picture, and most cases of lower rate of success due to tissue immersion procedure can be resolved with unmasking treatments.

### 3.4. The Importance of Investigating the Cerebral Cortex as a Place for Immature Neurons

As summarized in the Introduction, it has been recently shown that anatomical/physiological features linked to different mammalian species, such as brain size, gyrencephaly and lifespan, can be associated with differential amount/distribution of types of plasticity, likely due to evolutionary choices [7,20,22,23]. It is suggested that large-brained mammals might have reduced neurogenic capacity with respect to laboratory rodents [20,21,24,33], although this issue is still strongly debated [27,28]. On the other hand, a recent systematic, comparative study on layer II cortical immature neurons in twelve widely different mammals revealed a striking prevalence of these cells in large-brained species [23]. Of importance, in gyrencephalic mammals, including humans, these cortical immature neurons extend in the layer II-III of the entire neocortex [23,62,72], thus bringing the possibility of adding new neurons through a “neurogenesis without division” in brain regions endowed with high cognitive functions, yet devoid of stem cell-driven neurogenesis [11,15,16]. Since cortical immature neurons do not divide postnatally, they cannot be studied with cell proliferation markers used to characterize adult neurogenic processes, thus relying on markers for immaturity (e.g., DCX, PSA-NCAM [11]) or transgenic animals [8], the latter being limited to laboratory rodents. For this reason and considering the prevalence of these cells in large-brained mammals, it is important to trust the reliability of markers and antibodies in conditions that can substantially differ from those well-known and standardized in rodents.

In the present study, the canonical neurogenic regions (SVZ and SGZ) have been used as a positive control for cell proliferation and for the presence of well-characterized DCX+ neurons in the five animal species considered. In humans, only the cerebral cortex was analysed, our study having no interest in the assessment of persistent neurogenesis in humans, rather in testing the reliability of various antibodies across mammals. In this context, we think that controversies in DCX detection are linked to real interspecies differences rather than to technical issues, and data provided here on the cerebral cortex do support this view. Specifically for the cerebral cortex of primates, previous reports on DCX detection are controversial [45,61]. In a recent study conducted on macaques fixed by perfusion by using a series of immunocytochemical controls [45], it was suggested that DCX staining in neocortex might be far more restricted than previously thought. Like our results obtained in human neocortex, Liu et al. found a lack of staining with Santa Cruz (mouse) antibody in macaques, while the staining would always be specific (with all their antibodies) in the SVZ. Similar to our results obtained in marmoset neocortex, Liu et al. in macaques found non-specific staining in most neurons with the Abcam antibody. Again, with Santa Cruz (mouse) antibody, no signal is detectable in both macaques and marmosets (with the exception of SVZ). Only a difference emerged with our findings in marmosets, namely a positive signal in neocortex with Santa Cruz (goat) and cell signalling (rabbit) antibodies, while Liu et al. reported weak fluorescent staining in a few cells, a fact that might be explained by interspecies differences.

Overall, though the work by Liu et al. was conducted on macaques and our study considered marmosets and humans, the results obtained in neocortex indicate common traits in these primates, confirming that they can react differently from other non-primate species, and that DCX in the primate cortex can be lower than previously reported.

## 4. Materials and Methods

### 4.1. Brain Sample

Brains used in this study were collected from various institutions and tissue banks, all provided by the necessary authorizations. All experiments were conducted in accordance with current EU and Italian laws (for additional details on the animals used in this study see Table 6).

**Table 6 ijms-24-02514-t006:** Brain samples and antibodies used in this study.

Brain Tissues Analysed						
Species	Life Stage	Fixation	Fixative		PMI	Sample processing
Mouse	3 months	Perfusion	4% PFA		none	- Whole hemisphere cut into coronal slices (1–2 cm thick) and washed in phosphate buffer solution - Cryo-protection in sucrose solutions - Frozen by immersion in liquid nitrogen-chilled isopentane - Cut into 40 μm thick cryostat coronal sections
Marmoset	Adult	Immersion	4% PFA (15% picric acid)		1 h	
Rabbit	3.5 years	Perfusion	4% PFA		none	
Cat	1.5 years	Immersion	4% buffered formalin		1 h	
Sheep	2 years	Perfusion	4% PFA		none	
	5 years	Immersion	4% buffered formalin		20 min	
Human (see Table 7)	Adult (post-mortem)	Immersion	4% buffered formalin		16–23 h	
	Adult (intraoperative)		4% PFA		none	- Brain samples cut into slice (0.5 cm thick); then, processed as above
**Primary Antibodies Tested**						
**Antigen**	**Host**	**Type**	**Code**	**Raised against** (source: data sheet and company information)	**Dilution**	**Source**
DCX	Goat	Polyclonal	SC8066	Epitope within the last 50 c-terminal amino acids	1:1000	Santa Cruz Biotechnology, Dallas, Texas, USA
	Mouse	Monoclonal	SC271390	Amino acids 81-365 mapping at the C-terminus of Doublecortin of human origin		
	Rabbit	Polyclonal	ab18723	Synthetic peptide conjugated to KLH derived from within residues 300 to the C-terminus of human doublecortin		Abcam, Cambridge, UK
			4604	Antigenic sequence surrounds amino acid 350 tyrosine of human doublecortin		Cell Signalling, Danvers, Massachusetts, USA
	Guinea pig		ab2253	Epitope aminoacidic sequence: YLPLSLDDSDSLGDSM		Merck Millipore, Burlington, Massachusetts, USA
Ki67	Rabbit		ab15580	Synthetic peptide	1:500	Abcam, Cambridge, UK
	Mouse	Monoclonal	550609	Human Ki67		BD Pharmingen, San Diego, California, USA

Three young adult mice (*Mus musculus*) were analysed (for additional details on the animals used in this study see Table 6). Perfusion was performed under anaesthesia (i.p. injection of a mixture of ketamine, 100 mg/kg, Ketavet, Bayern, Leverkusen, Germany; xylazine, 5 mg/kg; Rompun, Bayer, Milan, Italy; authorization of the Italian Ministry of Health and the Bioethical Committee of the University of Turin; code 813/2018-PR, courtesy of Serena Bovetti) and brains were postfixed for 4 h.

Three young-adult female rabbits (*Oryctolagus cuniculus*) were used. Animals were deeply anesthetized (ketamine 100 mg/kg—Ketavet, and xylazine 33 mg/kg body weight—Rompun) and perfused with fixative (Italian Ministry of Health, authorization n. 66/99-A). Tissues were postfixed for 6 h.

Marmoset (*Callithrix jacchus*) brains were extracted 1 h after death and post-fixed for 3 months. The exact ages of the animals are unknown; they were aged as adults by experienced veterinarians.

Three young-adult cat (*Felis catus domestica*) brains were extracted post-mortem (the PMI was less than 1 h), fixed and kept in the fixative solution for a 1 month.

Three young-adult sheep (*Ovis aries*) brains were perfused through both carotid arteries with 2 L of 1% sodium nitrite in phosphate buffer saline, followed by 4 L of ice-cold 4% paraformaldehyde solution in 0.1 M phosphate buffer, pH 7.4. The brains were then dissected out, cut into blocks and postfixed in the same fixative for 48 h. Three young-adult sheep brains were extracted post-mortem (the PMI was less than 1 h), fixed and kept in fixative for 1 month.

Human (*Homo sapiens*) intraoperative brain samples were collected from the Neurosurgery Unit of the Humanitas Hospital during selected surgeries for brain tumour. A portion of healthy/perilesioned tissue of temporal lobe cortex resected for surgical reasons (considered free-of-diseases from clinical and pathological evaluation), were washed in cold NaCl 0.9% solution and directly (within 1–2 h) fixed in PFA 4% (ethical approvement by IRCCS Humanitas prot. Nr. 400/19). Three different donors were used for this study (Table 7).

Post-mortem human brain tissues were obtained from the Human Brain Collection Core (HBCC), National Institute of Mental Health Intramural Research Program (NIMH-IRP), with permission from the legal next-of-kin according to the National Institutes of Health Institutional Review Board and ethical guidelines under protocol 17-M-N073. Cases were obtained from the Offices of the Chief Medical Examiner of the District of Columbia. Clinical characterization, neuropathological screening, and toxicological analyses were performed as previously described [89,90] (Table 7).

**Table 7 ijms-24-02514-t007:** Origin of human samples.

**Intraoperative Tissues**					
**ID**	**Fixation**	**Age**	**Sex**	**PSI**	**Cause of Surgery**
226012	4% PFA	81	Male	<1 h	Diffuse high-grade glioma
201032	4% PFA	67	Female	<2 h	Gliosarcoma grade IV
130079	4% PFA	79	Male	<2 h	Glioblastoma grade IV
**Post-Mortem Tissues**					
**ID**	**Fixation**	**Age**	**Sex**	**PMI**	**Cause of Death**
1271	4% buffered formalin	46.6	Male	16 h	Thrombosis
1164	4% buffered formalin	48.1	Female	16 h	Homicide
1122	4% buffered formalin	48.6	Male	23 h	Cardiomegaly

### 4.2. Tissue Processing for Histology

The whole brain hemispheres were cut into coronal slabs (1–2 cm thick). The slabs were washed in a phosphate buffer (PB) 0.1 M solution, pH 7.4, for 24–72 h (on the basis of brain size) and then cryoprotected in sucrose solutions of gradually increasing concentration up to 30% in PB 0.1 M. Then, they were frozen by immersion in liquid nitrogen-chilled isopentane at −80 °C. Before sectioning, they were kept at −20 °C for at least 5 h (time depending on the basis of brain size) and then cut into 40 μm thick coronal sections using a cryostat. Free-floating sections were then collected and stored in cryoprotectant solution at −20 °C until staining.

Human intraoperative brain samples collected during surgery (2–4 cm size; Figure 3C″) were fixed on PFA 4% for 72–96 h (depending on sample size), washed in PBS 1X and maintained in 30% sucrose solution in PBS 1X at 4 °C. One single coronal slice (0.5 cm thick) was embedded in optimal cutting temperature compound and cut into 40 μm thick coronal sections by using a cryostat. Slices were mounted on SuperFrost plus slides (Fisher Scientific-Epredia™ SuperFrost Plus© Gold) and stored at −20 °C until use.

### 4.3. Comparable Neuroanatomy

Correspondent coronal brain sections were identified in the different animal species (mouse, marmoset, rabbit, cat, sheep) in order to include the four regions to be analysed: hippocampal dentate gyrus and forebrain subventricular zone (as internal positive controls for Ki67 antigen), paleocortex (piriform cortex) and parietal neocortex (as brain area hosting DCX+ non-newly born immature neurons in non-rodent species) (Figure 3A,B). Two cryostat sections, 40 µm thick, cut at different anterior–posterior coronal levels (Figure 3A,B) were analysed for each animal, for DCX and Ki67 antigen immunocytochemistry with all antibodies (Table 6).

Human intraoperative samples were cut out from the temporal lobe (Figure 3C′). Human post-mortem tissues were obtained from coronal slices (1 cm thick) including the parietal and temporal cortex (Figure 3C′).

### 4.4. Immunofluorescence Protocol

Due to the number of variables involved in this study (seven antibodies tested in four neuroanatomical regions obtained from brains of six widely different species), and due to our aim of testing the availability of different antibodies in different mammals, we performed all experiments under the same conditions, based on our previous experience (for antibody dilution, see Table 6). We only chose to perform an additional test with antigen retrieval (5 min citrate treatment, increased to 30 min for Ki67 antigen, see below), since dealing with tissues fixed in different conditions with respect to standard protocols are usually adopted in laboratory rodents. Our main goal, rather than finding the “perfect protocol” (which clearly depends on the animal species considered and other variables, such as the method of fixation used), was to define some basic conditions in which to test the substantial differences linked to animal species and antibodies.

For immunofluorescence staining, sections were rinsed in PBS 0.01 M, pH 7.4, then immersed in appropriate blocking solution (1–3% Bovine Serum Albumin, 2% Normal Donkey Serum, 1–2% Triton X-100 in 0.01M PBS, pH 7.4) for 90 min at RT. Then, the sections were incubated for 48 h at 4 °C with primary antibodies (see Table 6), and subsequently with appropriate solutions of secondary antibodies for 4 h at RT: cyanine 3 (Cy3)-conjugated anti-goat (1:400; Jackson ImmunoResearch, West Grove, PA-705-165-147), cyanine 3 (Cy3)-conjugated anti-rabbit (1:400; Jackson ImmunoResearch, West Grove, PA-711-165-152), cyanine 3 (Cy3)-conjugated anti-guinea pig (1:400; Jackson ImmunoResearch, West Grove, PA-706-165-148), cyanine 3 (Cy3)-conjugated anti-mouse (1:400; Jackson ImmunoResearch, West Grove, PA-715-165-150), Alexa 647-conjugated anti-mouse (1:400; Jackson ImmunoResearch, West Grove, PA-715-605-151), Alexa 647-conjugated anti-rabbit (1:400; Jackson ImmunoResearch, West Grove, PA 711-605-152). Immunostained sections were counterstained with 4′,6-diamidino-2-phenylindole (DAPI, 1:1000, KPL, Gaithersburg, MD, USA) and mounted with MOWIOL 4–88 (Calbiochem, Lajolla, CA, USA). All staining protocols were performed with and without antigen retrieval treatment, to exclude signal alteration due to a possible masking of the epitopes linked to fixation. When performed, antigen retrieval was performed using citrate buffer at 90 °C for 5 min (only for Ki67 nuclear staining of marmoset and sheep tissues fixed by immersion, it was extended to 30 min; Figure 6). Neither signal amplification nor tissue autofluorescence elimination protocols were applied to avoid signal alteration.

All antibodies used in this study were predicted to work on mouse brain tissue (as stated by their datasheet). Thus, to confirm the quality of our antibodies before conducting the experiment, this species was used as a control. All antibodies produced a staining consistent with previous data, in all regions of interest (see Section 2). Moreover, to exclude possible unspecific staining due to fluorescent secondary antibodies, primary antibody omission experiments were performed by replicating the immunofluorescence protocol (see above) without the primary antibody incubation. This resulted in a complete absence of staining.

### 4.5. RNAscope

Fixed frozen intraoperative human brain samples were sectioned coronally at 20 µm on a cryostat (Leica CM1860, Wetzlar, Germany) and mounted on SuperFrost Plus slides (Epredia, Breda, The Netherlands). RNA scope was performed following the purchaser instructions. Briefly, following washing in PBS 1X for 5 min, slides were baked 30 min at 60 °C in the HybEZ™ Oven and post-fixed using PFA 4% immersion for 3 h. Thereafter, sections were treated with ascending series of ethanol and left to dry after the last 100% ethanol step. Slices were pre-treated with hydrogen peroxide for 10 min, set at 98–102 °C in target retrieval solution for 10 min and treated with protease plus at 40 °C for 30 min in the HybEZ™ Oven. After pre-treatment, the single-plex, chromogenic RNAscope assay (RNAscope™ 2.5 HD Assay—RED Cat No. 322360) was performed using human DCX probe (Cat No. 489551), human Ubiquitine probe (Cat No. 31004) as positive control and DapB probe (Cat No. 310043) as negative control. Probes hybridization was performed at 40 °C for 2 h in the HybEZ™ Oven and then the signal was amplified using multiple amplifier probes. For the detection of signal, fast red chromogen was used.

Following RNAscope assay, immunofluorescence was performed to combine RNA and protein analysis. After the chromogen, the slides were washed in PBS-T (0.3% Triton in PBS 1X) and incubated in blocking solution (2% BSA and 0.25% Triton in PBS 1X) for 1 h. DCX (see Table 6) or GFAP (Cat No. ab53554; Abcam) primary antibody was incubated overnight at 4°. The following day the donkey anti-rabbit IgG and donkey anti-goat IgG secondary antibodies, Alexa Fluor™ (Thermofisher, Cat No. A31573; Cat No. A32849; respectively) and donkey anti-mouse IgG Secondary Antibody Alexa Fluor™ 647 (Thermofisher, Cat No. A-31571) was added for 1 h and the nuclei stained with DAPI (1:1000, KPL, Gaithersburg, MD, USA) [91,92].

### 4.6. Image Processing

Images were collected using either a Nikon Eclipse 90i confocal microscope (Nikon, Melville, NY, USA) or Zeiss Axio Observer confocal microscope Z1/7 (Zeiss, Oberkochen, Germany) or Leica TCS SP5 confocal microscope (Leica Microsystems CMS GmbH, Germany). To actually compare antibodies, the same acquisition settings were used in the whole study. Set acquisition parameters were adopted, based on previous experience with both antigens. Confocal settings: DCX: laser power (20%); gain (120); pinhole (30 µm); exposure time (1.68 µs). Ki67 antigen: laser power (10%); gain (130); pinhole (30 µm); exposure time (1.68 µs).

All images were processed using Adobe Photoshop CS4 (Adobe Systems, San Jose, CA, USA) and ImageJ version 1.53t (Wayne Rasband, Research Services Branch, National Institute of Mental Health, Bethesda, MD, USA). Adjustments to colour, contrast and brightness were avoided.

## 5. Conclusions

Overall, this study shows that substantial differences in the occurrence, quality and specificity of immunocytochemical staining for DCX and Ki67 antigen can exist when comparing different mammalian species whose brain tissues are treated with different antibodies. Real interspecies variation in the occurrence of antigens and high species specificity of different antibodies (and, to a lesser extent, regional differences) can impact even more than the type/degree of fixation. Our results confirm that special troubles and non-specific staining can be particularly frequent when studying DCX in the cerebral cortex of primates and humans using most of the currently available antibodies. This might be explained by the simple fact that most of the tools currently available have been developed for research in laboratory rodents. To overcome such an impasse, there is urgent need to develop antibodies tailored for, and tested on, large-brained species, including humans.

After the recent discovery that non-newly generated, “immature” or “dormant” neuronal populations can be present in non-neurogenic brain regions (cerebral cortex and amygdala), and that these neurons are particularly abundant and widespread in large-brained species, the need for reliable cell markers to correctly identify the “young” neurons across mammals is gaining more and more importance.

## Figures and Tables

**Figure 1 ijms-24-02514-f001:**
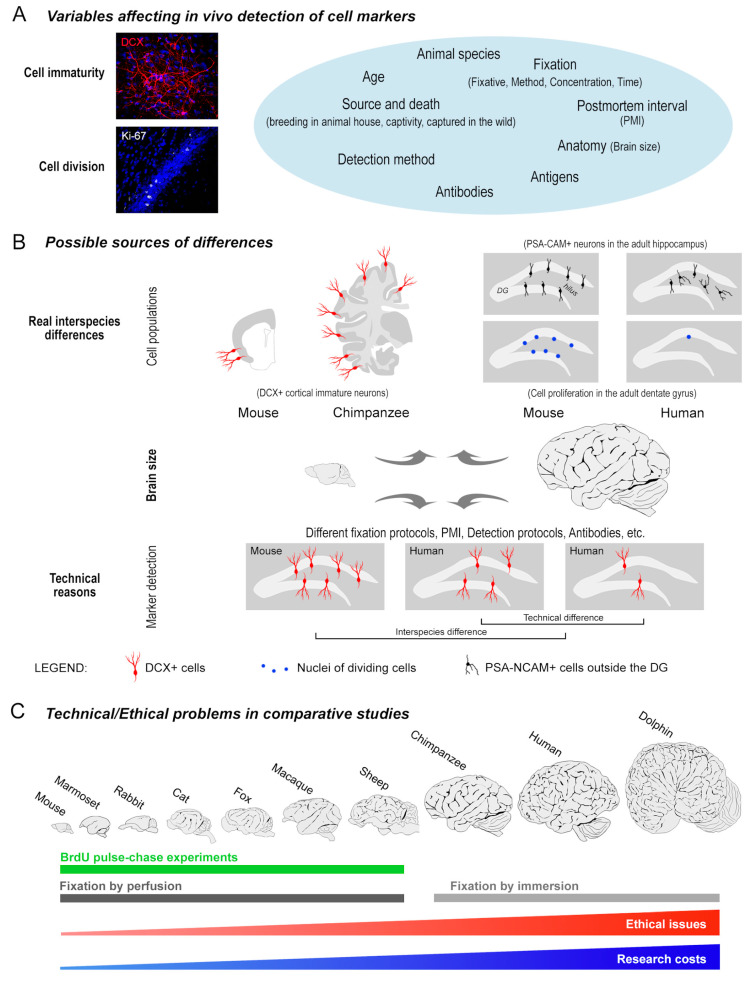
The complex landscape of cell marker detection in comparative neuroplasticity. (**A**) The two most used markers for in vivo detection of neuronal immaturity (doublecortin, DCX) and cell division (Ki67 antigen), and the main variables affecting their detection in the brain tissue; Image magnification: 40×. (**B**) The possible sources of variation can depend on several reasons, including differences linked to the animal species (neuroanatomy, evolutionary choices in the types of neuroplasticity, and intrinsic differences in the expression of cells/markers), and/or technical reasons linked to the procedures of brain sampling and fixation, potentially varying in relation to brain size. (**C**) Difficulties encountered in cell marker detection increase with increasing brain size and complexity of cognitive abilities (from mice to humans) on the basis of technical and ethical issues.

**Figure 2 ijms-24-02514-f002:**
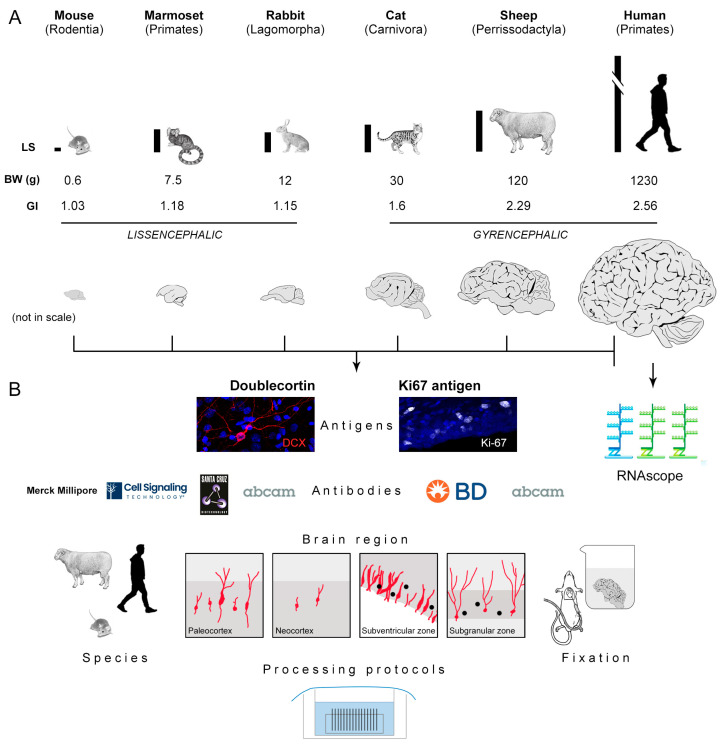
Experimental plan: (**A**) Animal species considered in this study, widely varying in brain size (BW: brain weight, in grams), gyrencephaly (GI: gyrification index) and lifespan (LS). (**B**) Schematic summary of the variables investigated; four brain regions are considered (squares: red cells, DCX; black dots, dividing nuclei). Logos reproduced with authorization of Companies.

**Figure 3 ijms-24-02514-f003:**
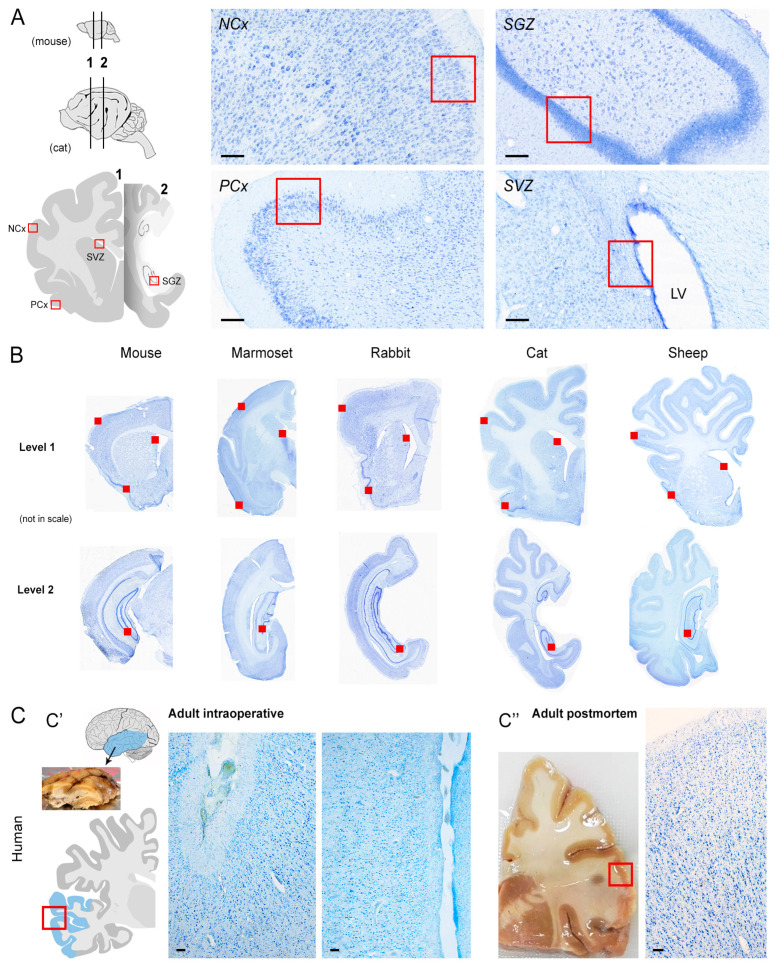
Anatomy of the brain regions studied. (**A**) Four different regions were considered (SVZ, subventricular zone; SGZ, sub-granular zone of the hippocampal dentate gyrus; NCx, neocortex; PCx, paleocortex; indicated by red boxes in (**A**,**C**), and red squares in (**B**)) at two different anterior–posterior, coronal levels (level 1: crossing the SVZ; level 2: crossing the hippocampus) in five mammalian species. (**B**) Whole coronal sections cut at the two brain levels, stained with toluidine blue and scanned with slidescanner Axioscan (Zaiss; Oberkochen, Germany). (**C**) Human brain tissues were obtained from intraoperative (**C′**) and post-mortem (**C″**) specimens corresponding to the temporal lobe (light blue). Scale bars: (**A**,**C**), 100 µm.

**Figure 4 ijms-24-02514-f004:**
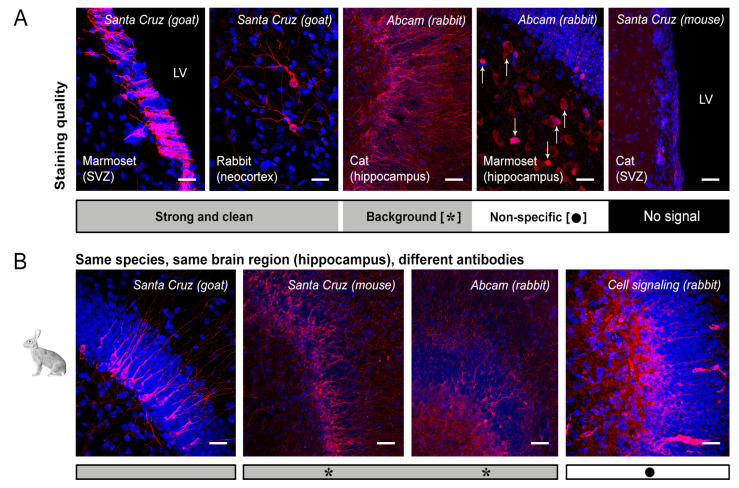
Type and quality of immunostaining obtained with different antibodies in different animal species and brain regions. (**A**) Four types of staining were considered, including a clear and clean staining without background noise, a specific staining bleary with background, an unspecific staining (with or without background; including artifacts, or staining associated with other structures, e.g., blood vessels, astrocytes, other neurons), and the absence of signal. Bottom: legend with symbols reported in the Results Tables. (**B**), Some examples considering substantial differences depending on the different antibodies used. LV: lateral ventricle. Scale bars: 30 µm.

**Figure 10 ijms-24-02514-f010:**
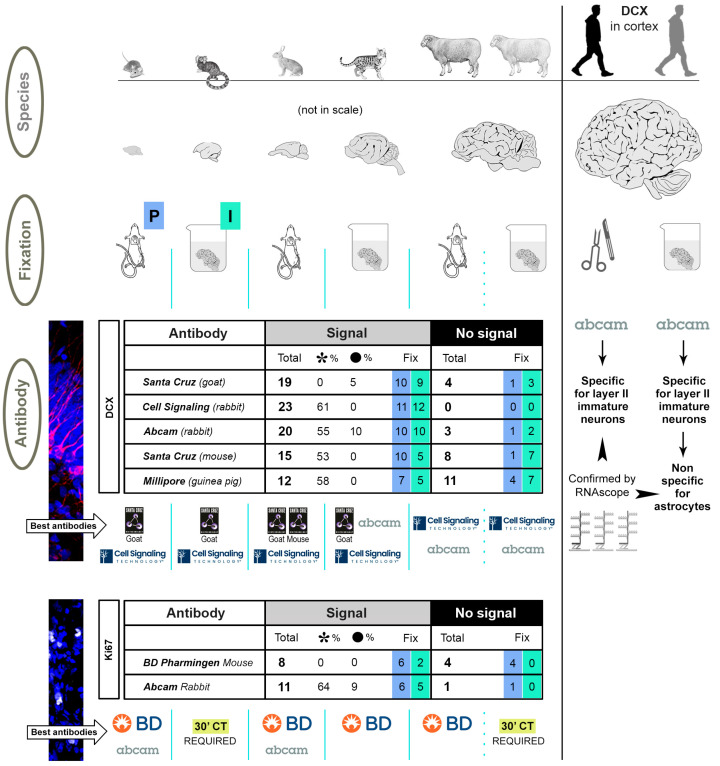
Schematic summary of the main results and conclusions. The three main variables considered in the study are reported in grey ovals on the left, including the animal species (characterized by widely different brain sizes), the type of fixation (perfusion: P, blue squares; immersion: I, green squares), and the commercial antibodies tested for DCX and Ki67 antigen. In the tables, the numbers of staining samples are reported (analysed in the four brain regions of the five mammalian species, each corresponding to two cryostat sections from three animals, treated with 5′ citrate), for a total of 115 staining samples for DCX (120 including the mouse neocortex lacking DCX+ cells), and 24 for Ki67. On the whole, a total of 89 staining samples for DCX made a positive signal, while 26 did not. For Ki67, there were 19 with positive staining, and 5 negative. The “No signal” column was considered to also include non-specific staining (i.e., non-successful staining). It is worth noting that the number of specimens fixed with perfusion or immersion (reported in blue and green squares, respectively) showing successful staining were almost equally distributed, especially for DCX. In the case of Ki67, we showed that some specimens fixed by immersion require longer time of citrate treatment to reveal staining (see Figure 6). The best-performing antibodies for each animal species are reported in tables below. Also in this case, the result is mostly independent from fixation (see for example the same outcome for DCX staining in perfused and immersed sheep brains), and rather linked to the association between animal species and antigen considered. In humans, only the Abcam antibody delivered a satisfactory result; on intraoperative specimens, the specificity of the staining was confirmed by RNAscope analysis (see Figure 9), also revealing a non-specific staining on astrocytes in the post-mortem tissue. Logos reproduced with authorization of Companies.

## Data Availability

The data of current study are available from the corresponding authors on reasonable request.

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
