# Peer review of "Consistency and Variation in Doublecortin and Ki67 Antigen Detection in the Brain Tissue of Different Mammals, including Humans"

_ijms, 2023, doi:10.3390/ijms24032514_

Round 1

Reviewer 1 Report

The authors reported in great detail evaluation methods for detection of immunohistochemical and mRNA expressions of doublecortin and Ki67 antigen in brain tissue of different mammals, including humans. These findings are very useful for researchers to perform immunohistochemical and mRNA evaluation of doubelcortin and Ki67 expressions in brain samples of mammals including humans. The reviewer has some questions.

1)      The authors described in Tables 6 and 7 that 4% buffered formalin was used as fixatives. Is it right? Was 4% buffered formalin used but not 10% buffered formalin?

2)      The authors reported in Table 3 the effects of citrate treatment for 5 minutes on immunostaining. What do the authors think about the effects of citrate treatment for 20 minutes or the effects of Tris-ethylenediaminetetraacetic acid (EDTA) treatment on immunostaining?

Author Response

The authors reported in great detail evaluation methods for detection of immunohistochemical and mRNA expressions of doublecortin and Ki67 antigen in brain tissue of different mammals, including humans. These findings are very useful for researchers to perform immunohistochemical and mRNA evaluation of doubelcortin and Ki67 expressions in brain samples of mammals including humans.

The Authors thank the Reviewer for appreciating our work.

The reviewer has some questions.

Query 1 (Q1)      The authors described in Tables 6 and 7 that 4% buffered formalin was used as fixatives. Is it right? Was 4% buffered formalin used but not 10% buffered formalin?

Response: yes, we confirm that tissues indicated in the Tables were fixed with 4% buffered formalin (4% formalin in PBS buffer; see also: Parolisi et al., 2015 Front Neuroanat, and Parolisi et al., 2017 Brain Struct Funct).

Usually, when indicated as 10% formalin, it is a 40% commercial formalin - the maximum concentration of the formaldehyde powder in solution - then used at 10%, thus resulting 4% in final concentration.

Q2)      The authors reported in Table 3 the effects of citrate treatment for 5 minutes on immunostaining. What do the authors think about the effects of citrate treatment for 20 minutes or the effects of Tris-ethylenediaminetetraacetic acid (EDTA) treatment on immunostaining?

Response: we decided to use 5 minutes of citrate treatment at 90°C based on our previous experience in regions and species analyzed in this study (e.g., Parolisi et al., 2017; Piumatti et al., 2018; La Rosa et al., 2018,2020). Given the general good quality of staining using this antigen retrieval method for DCX with different commercial antibodies (see Table 3b), we do not think that a longer time would ameliorate the staining pattern, whereas, as shown here, this would be useful for the nuclear staining of Ki67 antigen. To obtain the best (absolute or “perfect”) protocol in each case was not our main goal, also because it would depend on the animal species considered and/or the method of fixation used. Our goal was to define some basic conditions in which to test the substantial differences linked to animal species and antibodies. Then, once these differences are known, various treatments aimed at improving the staining can be applied in each single case, namely, by different laboratories working on different tissues.

A sentence was added in the Method section.

As for the Tris-EDTA treatment, it is worth noting that antigen retrieval in an acidic environment with citrate buffer was performed in most of the References listed in Table 1, except for two cases in which Tris-EDTA was used (Dennis et al., 2016; Perry et al., 2012), but only in human tissue. Thus, a good working experimental condition to unmask DCX antigen seems to be the one used in our study. Again, we defined some basic conditions in which to test the substantial differences linked to animal species and antibodies.

Reviewer 2 Report

Ghibaudi at al present a tedious and important comparison of DCX and Ki-67 immunostaining between 6 mammalian (both lissencephalic and gyrencephalic) species, using seven commercially available antibodies, and different fixation / antigen retrieval techniques. In addition, they provide a systematic and unbiased comparison of the staining patterns in non-human samples from 4 regions, while in case of human material, they compare intraoperative and postmortem cortical samples from the temporal lobe. Besides the 9 composite Figures, 5 informative Tables summarize the applied antibodies and the obtained results.

I consider their work as highly important, providing important technical data for any further investigations in the field of adult neurogenesis or identifying dormant, immature neurons. This is especially important in this field, as many of the current controversies are likely derived from inappropriate methodologies or not clarified interspecies differences. Altogether, I appreciated their work and have mainly minor comments.

* The Authors argue that in case of the human samples, they wanted to investigate only the temporal cortex – nevertheless, it would be also important to provide unbiased data using the different antibodies within the human SVZ and the DG, especially when considering the currently ongoing hot debate on adult human neurogenesis. Can the Authors supplement their human data with these samples?

* A general formatting remark: I think that some of the Figures have too small letterings, which are hard to read in the current version – please consider increasing the letter sizes (eg, Figure 1: “DCX+ cortical immature neurons”; Figure 2: latin names (could be even omitted) or “not in scale”; Figure 5 and 6: “perfusion” or “immersion”; Figure 8C: antibody types written over the images; Figure 9: tables summarizing the outcome of the results with the tested antibodies)

* Figure 4, Tables 3, 4 and 5: on one hand, it is good that the Authors tried to introduce symbols to describe the types of immunostaining – nevertheless, using four different backgrounds (white, grey, black and pink) in addition with white / black circles or asterisks is not easy to follow.

* I also suggest reducing the amount of the pictograms on Figure 9 – using “thumb up” or “thumb down” signs are not so appropriate in my opinion and some of the symbols (eg, the red outlines of the cells) are not even explained.

* In addition to my previous remark: there is no description at all how the images were taken and on which basis did they set the exposure times or additional settings. This is highly important to be able to separate “background” from the “strong and clean” staining types, not to mention the “no signal” category. Did they use the same image acquisition settings or did they optimize them according to the investigated samples? Please provide the necessary technical details in the revision.

* I appreciated the combination of the RNAscope and DCX immunostaining in case of human samples, nevertheless, Figure 8C images are too small to draw any conclusions, especially on co-labelling with the GFAP marker. Regarding the nature of the DCX+ neurons in Layer II: can the Authors identify the real identity of these cells?

* Did the Authors test different concentrations of the antibodies? This can also influence the differences in staining patterns.

* Please consider whether all self-citations are indeed necessary within the reference list (there are at least 20 from these).

Author Response

Ghibaudi at al present a tedious and important comparison of DCX and Ki-67 immunostaining between 6 mammalian (both lissencephalic and gyrencephalic) species, using seven commercially available antibodies, and different fixation / antigen retrieval techniques. In addition, they provide a systematic and unbiased comparison of the staining patterns in non-human samples from 4 regions, while in case of human material, they compare intraoperative and postmortem cortical samples from the temporal lobe. Besides the 9 composite Figures, 5 informative Tables summarize the applied antibodies and the obtained results.

I consider their work as highly important, providing important technical data for any further investigations in the field of adult neurogenesis or identifying dormant, immature neurons. This is especially important in this field, as many of the current controversies are likely derived from inappropriate methodologies or not clarified interspecies differences. Altogether, I appreciated their work and have mainly minor comments.

 The Authors thank the Reviewer for appreciating their work.

Q1) The Authors argue that in case of the human samples, they wanted to investigate only the temporal cortex – nevertheless, it would be also important to provide unbiased data using the different antibodies within the human SVZ and the DG, especially when considering the currently ongoing hot debate on adult human neurogenesis. Can the Authors supplement their human data with these samples?

Response: the present study was not intended to extend the analysis to human SVZ and SGZ, since the main objective, in humans, was directed to immature neurons present in the cerebral cortex. In this study the currently considered neurogenic sites (SVZ and SGZ) were used just as an internal control for both antigens in the five animal species. We highlighted this important point in the Discussion (see sentence in red). Moreover, we do not have large amounts of human brain tissue containing SVZ and SGZ, also because most work on humans, especially concerning the RNAscope analysis, was performed on intraoperative samples (cerebral cortex). For its complexity, a work dealing with human SGZ and SVZ might be a future study of its own.

A sentence was integrated in the Discussion, in order to make it clearer the aim of the study.

Anyway, we think that the current controversy concerning the existence of adult neurogenesis in the human hippocampus, is due to the interspecies differences in the occurrence/persistence of substantial cell division in the adult (an essential requirement to call it “adult neurogenesis”); indeed, different rates of such a division are detectable at different ages in mouse and humans. In our view, the presence of DCX+ cells in the adult hippocampus is not controversial; what is still not clear is their nature and origin (we think that most of them are not coming from neurogenesis but are “immature” neurons or a case of dematuration).

Q2) A general formatting remark: I think that some of the Figures have too small letterings, which are hard to read in the current version – please consider increasing the letter sizes (eg, Figure 1: “DCX+ cortical immature neurons”; Figure 2: latin names (could be even omitted) or “not in scale”; Figure 5 and 6: “perfusion” or “immersion”; Figure 8C: antibody types written over the images; Figure 9: tables summarizing the outcome of the results with the tested antibodies)

Response: we agree that in some complex figures the lettering was too small. We enlarged the lettering as requested, in Figure 1, 2, and 8. Tables in Figure 9 (now Figure 10) were also enlarged.

Latin names were deleted in Figure 2 (and reported in the Method section).

Q3) Figure 4, Tables 3, 4 and 5: on one hand, it is good that the Authors tried to introduce symbols to describe the types of immunostaining – nevertheless, using four different backgrounds (white, grey, black and pink) in addition with white / black circles or asterisks is not easy to follow.

Response: we thank the reviewer for outlining these points. We provided additional explanation in the Table legend, which has been expanded (NOTE: since this legend is not a text but a Tiff Figure, it is not highlighted in red).

It is also difficult to read the Tables within the template used for submission. This is due to the automatic line spacing, that cannot be reduced, so the Tables extend across many pages in the template. We tried to solve this formatting problem. Yet, whether Tables will not appear to be modified in the template, we will upload the PDF of the manuscript in which Tables are reduced in size, being contained in a single page.

Q4) I also suggest reducing the amount of the pictograms on Figure 9 – using “thumb up” or “thumb down” signs are not so appropriate in my opinion and some of the symbols (eg, the red outlines of the cells) are not even explained.

Response: we changed Figure 9 (now Figure 10 in the revised version) as requested, in order to make it clearer and more understandable. “Thumb up” or “thumb down” signs were removed. We hope that now the summary of Results/Conclusions provided by Figure 9 (now Figure 10) can be clear enough for the reader.

Q5) In addition to my previous remark: there is no description at all how the images were taken and on which basis did they set the exposure times or additional settings. This is highly important to be able to separate “background” from the “strong and clean” staining types, not to mention the “no signal” category. Did they use the same image acquisition settings or did they optimize them according to the investigated samples? Please provide the necessary technical details in the revision.

Response: of course, the acquisition settings used in our study (based on our previous experience with both antigens) were always the same, since optimizing the acquisition for each staining would add a variable which would make harder to compare all the different antibodies. Thus, no additional settings were used, in order to set the same conditions in the whole study.

We missed to make explicit this aspect in our first submission and now new information was provided in the Methods section.

Q6) I appreciated the combination of the RNAscope and DCX immunostaining in case of human samples, nevertheless, Figure 8C images are too small to draw any conclusions, especially on co-labelling with the GFAP marker. Regarding the nature of the DCX+ neurons in Layer II: can the Authors identify the real identity of these cells?

Response: we thank the Reviewer for the useful suggestion, and we split Figure 8 in two separate figures (now Figures 8 and 9). Revised Figure 8 includes data on immunocytochemistry and revised Figure 9 is dedicated to RNAscope. In Figure 9, enlarged images were provided, especially for the GFAP/RNAscope staining.

As to the real identity of the layer II DCX+ neurons, it remains unknown at present, even in mice. Experiments carried out with electrophysiology in a transgenic mouse allowing to visualize the DCX+ cells with green fluorescent protein (in the piriform cortex; Benedetti et al., 2019 Cereb Cortex) in order to follow their maturation, suggest that they are “principal neurons” of the layer II.

In humans, we know that only some of these cells do persist in adults, especially in the temporal cortex, appearing as immunoreactive cell bodies with scarce processes (likely because they are progressively losing their DCX expression, as a consequence of maturation (Li et al., 2022).

Q7) Did the Authors test different concentrations of the antibodies? This can also influence the differences in staining patterns.

Response: we based the antibody dilutions on our previous experience, where we tested anti-DCX and anti-Ki67 antibodies in both neurogenic sites and cerebral cortex of different mammals (e.g., Ponti et al., 2006; Parolisi et al., 2007; Piumatti et al., 2018; La Rosa et al., 2018,2020). In these previous reports we performed tests regarding optimal dilution.

In our experience, the antibody concentration can play a minor role with respect to the species-specific differences. In other words, an antibody can be working or not (or giving background or not) in the brain of a given animal species. When working, the dilution used in the present study has been proven to be appropriate.

In addition, this study reveals that in some cases a different staining (e.g., with or without background) can be obtained in different brain regions, in spite of the same antibody used at the same dilution.

A sentence was introduced in the Methods section and in the Discussion.

Q8) Please consider whether all self-citations are indeed necessary within the reference list (there are at least 20 from these).

Response: we are aware that the number of self-citations is high. This is because only a very restricted number of laboratories/research groups have studied or are dealing with cortical immature neurons, until now. Hence, most of the papers currently published on this topic do belong to these groups, and many of these publications come from our lab.

In addition, some of the self-citations are linked to Table 1 and 2, referring to the use of anti-DCX and anti-Ki67 antibodies in literature.

Based on the request, we tried to reduce the self-citations that can be considered not essential, and 4 of them were eliminated (listed below).

Bonfanti, L.; Charvet, C.J. Brain plasticity in humans and model systems: Advances, challenges, and future directions. Int. J. Mol. Sci. 2021, 22:9358. doi: 10.3390/ijms22179358

Parolisi, R.; Cozzi, B.; Bonfanti, L. Humans and dolphins: Decline and fall of adult neurogenesis. Front. Neurosci. 2018, 12, 497. doi: 10.3389/fnins.2018.00497

Parolisi, R.; Peruffo, A.; Messina, S.; Panin, M.; Montelli, S.; Giurisato, M.; Cozzi, B.; Bonfanti, L. Forebrain neuroanatomy of the neonatal and juvenile dolphin (T. truncatus and S. coeruloalba). Front. Neuroanat. 2015, 9:140. doi: 10.3389/fnana.2015.00140

La Rosa, C.; Bonfanti, L. Brain plasticity in mammals: An example for the role of comparative medicine in the neurosciences. Front. Vet. Sci. 2018, 5, 274. doi: 10.3389/fvets.2018.00274

Note: within the general request to add some technical details, additional information was provided in Table 6 concerning the epitopes recognized by the commercial antibodies (on the basis of the information available from data sheet or directly provided by the companies).